# Towards Efficient and Accurate Winograd Convolution via Full Quantization

**Tianqi Chen**[1,2], **Weixiang Xu**[1], **Weihan Chen**[1], **Peisong Wang**[1,2,3][†] **Jian Cheng**[1,2,3,4][†]

[1]Institute of Automation, Chinese Academy of Sciences
[2]School of Artificial Intelligence, University of Chinese Academy of Sciences
[3]AIRIA [4]Maicro.ai
{chentianqi2023, xuweixiang2018,chenweihan2018}@ia.ac.cn,
{peisong.wang,jcheng}@nlpr.ia.ac.cn

## Abstract

The Winograd algorithm is an efficient convolution implementation, which performs calculations in the transformed domain. To further improve the computation efficiency, recent works propose to combine it with model quantization. Although Post-Training Quantization has the advantage of low computational cost and has been successfully applied in many other scenarios, a severe accuracy drop exists when utilizing it in Winograd convolution. Besides, despite the Winograd algorithm consisting of four stages, most existing methods only quantize the element-wise multiplication stage, leaving a considerable portion of calculations in full precision. In this paper, observing the inconsistency among different transformation procedures, we present PTQ-Aware Winograd (PAW) to optimize them collaboratively under a unified objective function. Moreover, we explore the full quantization of faster Winograd (tile size $\geq 4$) for the first time. We further propose a hardware-friendly method called Factorized Scale Quantization (FSQ), which can effectively balance the significant range differences in the Winograd domain. Experiments demonstrate the effectiveness of our method, e.g., with 8-bit quantization and a tile size of 6, our method outperforms the previous Winograd PTQ method by 8.27% and 5.38% in terms of the top-1 accuracy on ResNet-18 and ResNet-34, respectively.

## 1   Introduction

Recently, Convolution Neural Networks (CNNs) have demonstrated state-of-the-art performance in various computer vision tasks [1; 2; 3; 4]. However, the high computation and storage cost hinders their deployment on resource-limited devices. To address this problem, various solutions have been proposed in the literature, including network pruning [5], low-rank decomposition [6], network quantization [7; 8; 9; 10; 11] and faster convolution implementation [12; 13]. In this paper, we focus on network quantization, faster convolution implementation, and their combination.

Model quantization converts the floating-point weights and activations to low-bit integers. According to whether data with labels are required for training, quantization can be divided into two categories: Quantization-Aware Training (QAT) [14; 15; 16; 17] and Post-Training Quantization (PTQ) [18; 19; 20; 21]. Although QAT can achieve promising performance, training the network requires many GPU resources and a full training dataset. Therefore, it is not always practical when either the training dataset is unavailable (e.g., privacy and commercial training data) or rapid deployment is required. On the contrary, as PTQ only needs few-shot, unlabeled calibration data and fewer computation resources, it is widely applicable in the industry.

---

[†]Corresponding authors.

37th Conference on Neural Information Processing Systems (NeurIPS 2023).

An alternative method for enhancing the speed of CNNs involves the development of faster convolution implementations, such as FFT [13] and Winograd [12]. Among them, the Winograd algorithm is the most popular fast convolution operator [22]. Recent research [23; 24; 25] has focused on combining the Winograd algorithm with model quantization to further utilize the benefits of both techniques. However, two significant problems remain: On the one hand, while Winograd convolution with larger tile sizes (tile size$\geq$4) can provide more significant acceleration, existing PTQ methods for such convolution still suffer from drastic accuracy degradation. On the other hand, most existing works [24; 26] only quantize the element-wise multiplications, while leaving the domain transformation implemented by full-precision matrix multiplications, which can comprise a significant portion of the computation (40.1% when the input and output channels are 32 [24]). However, according to our experiments (Table 2 and Figure 2), these matrix multiplications are challenging to quantize because of the significant distribution imbalance in the Winograd domain.

In this paper, we first identify that quantization makes the transformation procedures in the Winograd algorithm inconsistent. Based on this observation, we propose **P**TQ-**A**ware **W**inograd (PAW), which optimizes transformation matrices collaboratively under a unified objective function. We then explore the fully quantized Winograd convolution, which is the first time for large tile sizes. We empirically find it is a non-trivial task because of the significant range differences of pixels in the Winograd domain. Through theoretical analysis of this phenomenon, we discover that it is possible to achieve comparable performance to per-pixel quantization without sacrificing computational efficiency via Factorized Scale Quantization (FSQ), which factorizes the tile size scales into vector size scales. Experiments are conducted to demonstrate the effectiveness of our method, e.g., with 8-bit quantization and a tile size of 6, our method outperforms the previous Winograd PTQ method by 8.27% and 5.38% in terms of the top-1 accuracy on ResNet-18 and ResNet-34, respectively.

Overall, our contributions in this work are threefold:

- We experimentally observe that quantization will disrupt the consistency between different transformation procedures and propose PTQ-Aware Winograd, which utilizes a unified optimization procedure to make the Winograd algorithm more robust to quantization.

- Through extensive experiments and theoretical analysis, we propose Factorized Scale Quantization, a hardware-friendly method suitable for the distribution characteristic of the Winograd domain tensors.

- To the best of our knowledge, we are the first to achieve full quantization of Winograd convolution with large tile sizes (4 and 6). Experiments prove that our proposed method shows significant improvements over previous PTQ methods, even under fully-quantization settings.

## 2   Related Works

**Quantization-aware training (QAT)** [8; 27; 28; 29; 30] is a technique that simulates quantization noise during end-to-end training of neural networks. It uses discretized weights during both forward and backward propagation and updates original full-precision weights. Since the gradient of the quantization function is either zero or undefined everywhere, this procedure is done via a gradient approximation method called straight-through estimator [31]. Recently, some methods also add quantization related parameters to this training procedure, such as clipping range [32] and step size [14]. Although QAT methods have promising performance, they need the whole dataset and huge GPU resources.

**Post-training quantization (PTQ)** [7; 21; 9; 18; 33] is a more lightweight method that does not require retraining the network end-to-end. It only needs a small number of samples to estimate activation distribution. Despite its efficiency, PTQ suffers from a more significant accuracy degradation than QAT. The research community has actively proposed various methods to alleviate this problem. For example, [21] observes the scaling equivariance of activation functions and proposes to balance weights in consecutive layers. They also propose bias correction, which absorbs high biases into the next layer. Recently, some methods have found that the change in feature maps is a practical proxy signal for final task loss and use a layer-wise feature map reconstruction procedure to boost performance. Bit-Split [7] turns the low-bit discrete optimization into multiple single-bit optimization problems to find the optimal quantization scales and integer weights. AdaRound [18]

Table 1: Percentage of the computational burden of different operations in F(6,3) Winograd convolution. Experiments are conducted on different blocks of ResNet-20. "Offline" means this operation can be preprocessed before inference.

| Operation | Block 0 | Block 1 | Block 2 | Total |
|---|---|---|---|---|
| $B^T X B$ | 22% | 16% | 10% | 16% |
| $U \odot V$ | 39% | 56% | 72% | 56% |
| $A^T O A$ | 39% | 28% | 18% | 28% |
| $GWG^T$ | Offline | Offline | Offline | Offline |

proposes optimizing the round function with a soft relaxation, which makes huge progress than the original round-to-nearest function.

**Winograd convolution quantization** Winograd is an fast convolution implementation first applied by [12]. To further improve computational efficiency, many works [23; 24; 34; 35; 36] focus on combining it with model quantization. For Winograd convolution networks with tile size $\geq 4$, most works can only achieve an acceptable accuracy drop when using QAT, where the strong effectiveness of retraining the whole network hides the difficulty of quantization [23; 34; 36]. For instance, BQW [24] observes the heterogeneity of channel ranges in transformed tensors and equalizes them via migrating the scale variance from activations to weights. However, compared to the significant performance improvements achieved in QAT, their method only achieves limited progress in the case of PTQ. One of the reasons, we think, is that their method ignores the quantization impact on the Winograd algorithm, which is more notable without fine-tuning the whole network. In this paper, we will solve this problem and explore a more challenging scenario, **fully quantized Winograd convolution using PTQ**.

## 3 Background

### 3.1 Winograd

Winograd proposes the minimum filtering algorithm of finite impulse response (FIR) filtering in [37]. For $r \times r$ standard convolutions with filter size $r$, the algorithm transforms the convolution operations to the Winograd domain and generates $m \times m$ (spatial) outputs at a time, which is denoted as $F(m, r)$. The parameter $m$ is called tile size, which is used to balance the speedup and numerical precision. The Winograd algorithm can be divided into four stages:

**Input transformation**: Firstly, a patch $X \in \mathbb{R}^{C_i \times a \times a}$ is extracted from the input data, with patch size $a = m + r - 1$. Then the c-th channel sub-tensor of $X$, denoted as $X_c$, is transformed into the Winograd domain using input transformation matrix $B \in \mathbb{R}^{a \times a}$. In this paper, the transformed inputs are denoted as $U \in \mathbb{R}^{C_i \times a \times a}$:

$$U_c = B^T X_c B, \quad c = 1, ..., C_i \tag{1}$$

**Weight transformation**: Similarly, weights $W \in \mathbb{R}^{C_o \times C_i \times r \times r}$ are transformed into the Winograd domain to get transformed weights $V \in \mathbb{R}^{C_o \times C_i \times a \times a}$. This process can be performed offline before inference because model weights are frozen after training.

$$V_{f,c} = GW_{f,c}G^T, \quad G \in \mathbb{R}^{a \times r} \quad \text{where} \quad f = 1, ..., C_o, \quad c = 1, ..., C_i \tag{2}$$

**Element-wise multiplication**: In the Winograd domain, the convolution operation is performed by element-wise multiplications between $U$ and $V$:

$$O_f = \Sigma_c^{C_i} U_c \odot V_{f,c} \tag{3}$$

where $\odot$ denotes element-wise multiplication.

**Output transformation**: Finally, $O \in \mathbb{R}^{C_o \times a \times a}$ are transformed back to the feature map domain, and then, we get final outputs $Y \in \mathbb{R}^{C_o \times m \times m}$:

$$Y_f = A^T O_f A, \quad A \in \mathbb{R}^{a \times m} \quad \text{and} \quad f = 1, ..., C_o \tag{4}$$

Depending on the particular choice of Winograd domain (i.e., polynomial domain), these transformation matrices $A$, $B$, and $G$ in Eq. (1)-(4) can be different. For F(2,3), the most common choice of the Winograd domain is $f(x) = (x-1)(x+1)x$, then $A$, $B$ and $G$ can be constructed as follows[12]. More details are provided in the supplemental materials.

$$A^T = \begin{bmatrix} 1 & 1 & 1 & 0 \\ 0 & 1 & -1 & -1 \end{bmatrix}, \quad B^T = \begin{bmatrix} 1 & 0 & -1 & 0 \\ 0 & 1 & 1 & 0 \\ 0 & -1 & 1 & 0 \\ 0 & 1 & 0 & -1 \end{bmatrix}, \quad G = \begin{bmatrix} 1 & 0 & 0 \\ \frac{1}{2} & \frac{1}{2} & \frac{1}{2} \\ \frac{1}{2} & -\frac{1}{2} & \frac{1}{2} \\ 0 & 0 & 1 \end{bmatrix} \quad (5)$$

Although the Winograd algorithm reduces the computational complexity of convolution operations by performing them in the Winograd domain, these transformation steps also bring additional overheads. An example is shown in Table 1. When the channel numbers $C_o$ and $C_i$ are not large enough, these transformation processes will incur considerable computational costs.

### 3.2 Quantization

Quantization converts a full-precision tensor to an integer one. To improve hardware simplicity and efficiency, we utilize symmetric uniform quantization without zero-point. For a tensor $V$, quantization maps it to the integer values $\widetilde{V}$ and de-quantization remaps it to float-point values $Q(V)$:

$$\widetilde{V} = \left\lfloor clip(\frac{V}{s}, -q_{min}, q_{max}) \right\rceil \quad (6)$$

$$Q(V) = \widetilde{V} \cdot s \quad (7)$$

where $s$ represents a full-precision scalar called quantization scale, $\lfloor z \rceil$ rounds $z$ to the nearest integer and $clip(z, r_1, r_2)$ clamps $z$ into the range $[r_1, r_2]$. For symmetric quantization with $B$ bits, $q_{min} = -2^{B-1}$ and $q_{max} = 2^{B-1}$.

In Eq. (6), we utilize a scalar $s$ to quantize the entire tensor, called per-tensor quantization. Previous studies [24; 26] have shown that using per-pixel quantization, which provides independent scales for different pixels of transformed tensors, can significantly improve performance:

$$\widetilde{U}_c = \lfloor U_c \oslash S_U \rceil \quad (8)$$

and

$$\widetilde{V}_{f,c} = \lfloor V_{f,c} \oslash S_V \rceil \quad (9)$$

Here $\oslash$ is the element-wise division and we emit the $clip$ function. $S_U$ and $S_V \in \mathbb{R}^{a \times a}$ are tile size scales and can be factored out of the summation of Eq. (3). Thus, we can perform MAC operations in fixed-point format:

$$O_f = \underbrace{(\Sigma_c^{C_i} \widetilde{U}_c \odot \widetilde{V}_{f,c})}_{int8} \odot (S_U \odot S_V) \quad (10)$$

## 4  Method

### 4.1  PTQ-aware Winograd algorithm

While current state-of-the-art PTQ methods [18; 19; 9; 33] can achieve near-lossless 6-bit quantization on popular non-Winograd CNNs, recent research has shown that 8-bit quantization of Winograd convolutions still suffers from a significant accuracy drop (e.g., 9.71% for F(6,3) on ResNet-18 [24]). We argue that one key reason why quantization and the Winograd algorithm are not well-compatible is that quantization alters input and weight transformation procedures, making the whole algorithm inconsistent.

To support our argument, we experiment on the first layer of ResNet-18 and take weight transformation as an example. After quantization, the weight transformation procedure is changed from $GWG^T$ to $Q(GWG^T)$. Consequently, the other two transformation procedures, $A^TOA$ and $B^TXB$, will not be consistent with this new weight transformation. To illustrate it, we randomly perturb $A$ and $B$ and compute the reconstruction loss $\mathcal{L}_{quant}$ and $\mathcal{L}_{origin}$ with or without quantization, respectively:

$$\mathcal{L}_{origin} = \Sigma_f^{C_o} ||A^T(\Sigma_c^{C_i}(B^T X_c B) \odot (GW_{f,c}G^T))A - Y_f||^2 \quad (11)$$

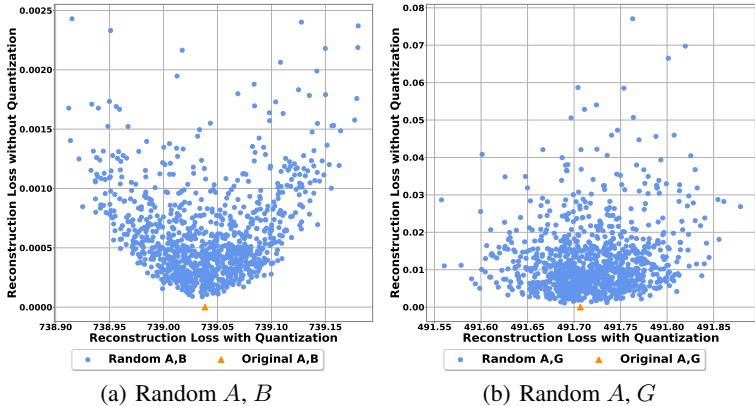

(a) Random $A$, $B$                      (b) Random $A$, $G$

Figure 1: Reconstruction loss points obtained by randomly perturbing $A$, $B$ (a) or $A$, $G$ (b). The bottom points indicate a minor loss without quantization. And the left points indicate a minor loss with quantization.

$$\mathcal{L}_{quant} = \Sigma_f^{C_o}||A^T(\Sigma_c^{C_i}(B^T X_c B) \odot Q(GW_{f,c}G^T))A - Y_f||^2 \tag{12}$$

The results are shown in Figure 1(a). Although original $A$ and $B$ can produce desired results with floating-point $GWG^T$ (the yellow triangle on the bottom), they are no longer optimal when changing $GWG^T$ to $Q(GWG^T)$. Concretely, about half of the random perturbed matrices lead to smaller errors (the blue dots on the upper left of the yellow triangle). A similar phenomenon also exists for input transformation, as shown in Figure 1(b). These experiments indicate that the inconsistency between domain transformations caused by quantization renders the original matrices unsuitable for quantization scenarios, which is the primary reason why Winograd convolutions are difficult to quantize. In order to align these transformation procedures after quantization, we propose to adjust transformation matrices via an optimization procedure as follows:

$$\underset{A,B,G}{argmin} \quad \mathbb{E}_{X \sim \mathcal{D}} \left[ \Sigma_f^{C_o}||A^T(\Sigma_c^{C_i}Q(B^T X_c B) \odot Q(GW_{f,c}G^T))A - Y_f||^2 \right] \tag{13}$$

Since the transformation procedures in the Winograd algorithm are adopted considering the effect of PTQ, we name this method as **PTQ-Aware Winograd** (PAW). By using the straight-through estimator [31] to approximate the gradient through the round function as a pass-through operation, we can obtain the derivative of $G$ (derivatives of $A$ and $B$ are shown in supplemental materials) and update it using SGD:

$$\frac{\partial \mathcal{L}}{\partial G} = \Sigma_f^{C_o}\Sigma_c^{C_i}(\frac{\partial L}{\partial O_f} \odot Q(U_c))GW_{f,c}^T + (\frac{\partial L}{\partial O_f} \odot Q(U_c))^T GW_{f,c} \tag{14}$$

where

$$\frac{\partial L}{\partial O_f} = 2A(A^T O_f A - Y)A^T \tag{15}$$

### 4.2 Fully-quantized Winograd convolution

#### 4.2.1 Motivation

Winograd convolution involves a sequence of computational steps from Eq. (1) to Eq. (4). To further improve computation efficiency via full quantization, all these operations need to be quantized except

Table 2: Exploration on 8-bit quantization of different components of Winograd convolution on ResNet-18. Output tensor in the Winograd domain ($O$) is the primary cause of accuracy degeneration.

|  | FP | $A$ | $B$ | $X$ | $U$ | $V$ | $O$ | $B^T X$ | $A^T O$ |
|---|---|---|---|---|---|---|---|---|---|
| Accuracy | 69.76 | 69.74 | 69.66 | 69.67 | 69.40 | 69.40 | **0.20** | 69.22 | 66.56 |

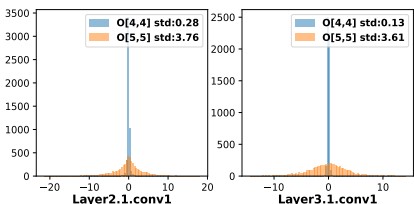

Figure 2: The ranges of $O$ vary widely between different pixels. In both layers, the std of $O_{5,5}$ is ten times larger than that of $O_{4,4}$.

Table 3: The performance of different quantization schemes for $O$. Although per-pixel quantization is excellent in maintaining model accuracy, it is not supported in hardware.

| Granularity | Accuracy | Hard-friendly |
|---|---|---|
| Quantize $O$ (per-tensor) | 0.216 | ✓ |
| Quantize $O$ (per-pixel) | 68.740 | ✗ |
| Quantize $O$ (ours) | 67.866 | ✓ |

Eq. (2), which can be done offline. However, we empirically find that it is non-trivial to achieve full quantization because that accuracy will drop to zero due to the quantization of $O$ (i.e., the output tensor in the Winograd domain), as shown in Table 2. After further investigation, we find that the ranges of $O$ vary widely between different pixels. For example, as shown in Figure 2, the standard deviation (std) of $O$ distributed at pixel (4,4) is ten times larger than that at pixel (5,5). A shared quantization scale for them is difficult to achieve both small round error and clamping error. A possible solution is to provide independent scales for them, i.e., to use per-pixel quantization (Table 3). However, since $O$ will take part in matrix multiplication (Eq. 4) instead of element-wise multiplication, per-pixel quantization will lead to different scales in the summation dimension, which makes it not feasible in general hardware.

### 4.2.2 Factorized-scale quantization

Motivated by the strong range difference, we present the following theorem, which shows the cause of this distribution characterization and proves the standard deviation of $O$ can be factorized into two vectors.

**Proposition 1.** *Assume all elements of $W \in \mathbb{R}^{C_i \times r \times r}$ and $X \in \mathbb{R}^{C_i \times a \times a}$ are independently and identically distributed variables with zero mean (e.g., $X \sim \mathcal{N}(0, \sigma_X)$, $W \sim \mathcal{N}(0, \sigma_W)$). If denote $B^T = (b_1^T, b_2^T, ..., b_a^T)^T$ and $G = (g_1^T, g_2^T, ..., g_a^T)^T$, we have:*

$$Var\left[O_{ij}\right] = Var\left[\Sigma_c^{C_i}(B^T X_c B \odot GWG^T)_{ij}\right] = u_i v_j \tag{16}$$

*where $u_i = \sqrt{C_i \sigma_X \sigma_W} ||b_i||^2 ||g_i||^2$ and $v_j = \sqrt{C_i \sigma_X \sigma_W} ||b_j||^2 ||g_j^2||$.*

*Proof.* Firstly, since $B^T X_c B$ and $GW_c G^T$ are the linear combinations of independent and identical variables with zero mean, we can calculate their mean and variance:

$$E\left[(B^T X_c B)_{ij}\right] = 0 \quad and \quad E\left[(GW_c G^T)_{ij}\right] = 0, \tag{17}$$

$$Var\left[(B^T X_c B)_{ij}\right] = Var\left[b_i^T X_c b_j\right] = ||b_i||^2 ||b_j^2|| \sigma_X, \tag{18}$$

$$Var\left[(GW_c G^T)_{ij}\right] = Var\left[g_i^T W_c g_j\right] = ||g_i||^2 ||g_j^2|| \sigma_W \tag{19}$$

Then we can derive the variance of $O$:

$$Var\left[O_{ij}\right] \overset{(a)}{=} \Sigma_c^{C_i} Var\left[(B^T X_c B)_{ij} \cdot (GW_c G^T)_{ij}\right]$$
$$\overset{(b)}{=} \Sigma_c^{C_i} Var\left[(B^T X_c B)_{ij}\right] Var\left[(GW_c G^T)_{ij}\right] = C_i ||b_i||^2 ||b_j||^2 ||g_i||^2 ||g_j^2|| \sigma_X \sigma_W \tag{20}$$

The equation (a) holds because the variables in different channels are independent. The equation (b) holds because the random variables $U$ and $V$ are independent of each other and have zero mean. $\square$

After finding that the standard deviation of $O$ can be decoupled along rows and columns separately, an intuitive thought is that the per-pixel scales may also satisfy similar rules. A more theoretical proof is provided as Theorem 1.

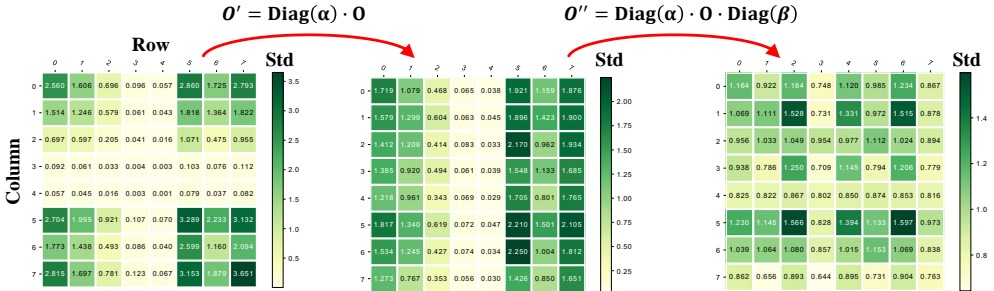

Figure 3: An illustration of Factorized Scale Quantization. Since the standard deviation of $O$ can be decoupled along rows and columns separately, using two vector scales, $\alpha$ and $\beta$, is enough for balancing $O$ to a similar distribution.

**Theorem 1.** *Assuming $X \sim \mathcal{N}(0, \sigma_X)$, $W \sim \mathcal{N}(0, \sigma_W)$, the optimal per-pixel scales $S^*$ which minimize quantization error in Eq.* (21) *can be factorized as: $S^* = \alpha * \beta^T$.*

$$\underset{S}{argmin} \quad E_{X \sim \mathcal{N}(0,\sigma_X), W \sim \mathcal{N}(0,\sigma_W)} \left[ ||O - Q_S(O)||^2 \right] \tag{21}$$

*Proof.* In Proposition 1, we have proven that:

$$Var\left[O_{ij}\right] = \sqrt{C_i \sigma_X \sigma_W}||b_i||^2||g_i||^2 \cdot \sqrt{C_i \sigma_X \sigma_W}||b_j||^2||g_j^2|| \tag{22}$$

Observing that $O_{ij}$ is the summation of $C_i$ independent variables, where $C_i$ is the number of input channels and is usually large (e.g., 128). According to the Central Limit Theorem [38], we can assume $O_{ij}$ follows Gaussian distribution. In supplementary material, we will show that the optimal scale $s$ to minimize the mean-square error of quantizing Gaussian variables $z \sim \mathcal{N}(0, \sigma^2)$ is proportional to $\sigma$, i.e., $s = K\sigma$, where $K$ is a constant. Thus, we have:

$$S_{ij} = \sqrt{KC_i||b_i||^2||g_i||^2\sigma_X\sigma_W} \cdot \sqrt{KC_i||b_j||^2||g_j^2||\sigma_X\sigma_W} = \alpha_i \cdot \beta_j \tag{23}$$

$\square$

The benefit of factorizing per-pixel scales into two vectors is that we can move both scales into transformation matrices. So we can facilitate per-tensor matrix multiplication implementation when per-pixel quantization is utilized:

$$
\begin{aligned}
A^T O A &\approx A^T \cdot (S_O \odot \widetilde{O}) \cdot A \\
&= A^T \cdot ((\alpha * \beta^T) \odot \widetilde{O}) \cdot A \\
&= A_L \widetilde{O} A_R
\end{aligned} \tag{24}
$$

where:

$$A_L = A^T \cdot Diag(\alpha), \quad A_R = Diag(\beta) \cdot A \quad and \quad \widetilde{O} = \lfloor O \oslash S_O \rceil \tag{25}$$

Note that at this stage, matrix $A$ is not quantized. After rescaling, we can further quantize these transformation matrices and middle results to obtain fully-quantized Winograd convolution. In this paper, this method is called **F**actorized **S**cale **Q**uantization (FSQ) and an illustration of it is shown in Figure 3.

### 4.2.3 Optimization procedure

In this section, we focus on how to determine the optimal $\alpha$ and $\beta$. Given $N$ samples $\{O^1, O^2, ..., O^N\}$, we can minimize the quantization error via the following non-linear least square regression problems:

$$\underset{\alpha, \beta}{argmin} \Sigma_n^N \Sigma_f^{C_o} ||O_f^n - (\alpha * \beta^T) \odot \widetilde{O_f^n}||^2 \tag{26}$$

The optimization of Eq. (26) is non-trivial due to the integer constraint of $\widetilde{O}$. Thus we propose an iteration method to solve it.

**Solving $\alpha$ when fixed $\beta$ and $\widetilde{O}$.** When $\beta$ and $\widetilde{O}$ are fixed, (26) is a quadratic function with respect to $\alpha$. Then the closed-form solution for $\alpha$ can be easily obtained as:

$$\alpha^* = \frac{\Sigma_n^N \Sigma_f^{C_o} (O_f^n \odot \widetilde{O}_f^n) \cdot \beta}{\Sigma_n^N \Sigma_f^{C_o} (\widetilde{O}_f^n \odot \widetilde{O}_f^n) \cdot (\beta \odot \beta)} \tag{27}$$

**Solving $\beta$ When fixed $\alpha$ and $\widetilde{O}$.** The closed-form solution for $\beta$ is similar to that of $\alpha$:

$$\beta^* = \frac{\Sigma_n^N \Sigma_f^{C_o} (O_f^n \odot \widetilde{O}_f^n)^T \cdot \alpha}{\Sigma_n^N \Sigma_f^{C_o} (\widetilde{O}_f^n \odot \widetilde{O}_f^n)^T \cdot (\alpha \odot \alpha)} \tag{28}$$

**Solving $\widetilde{O}$ when fixed $\alpha$ and $\beta$.** When $\alpha$ and $\beta$ are fixed, $\widetilde{O}$ is simply the integer values by rounding to the nearest:

$$\widetilde{O}_f^n* = \left\lfloor O_f^n \oslash (\alpha^T \cdot \beta) \right\rceil \tag{29}$$

## 5 Experiments

### 5.1 Experimental settings

We use full-precision pre-trained models and replace all 3x3 convolutions (stride 1) with Winograd convolutions. All convolutions, including the first and last layers, are quantized using symmetric quantization. Following BQW[24], we use per-pixel quantization for transformed inputs $U$ and weights $V$. In order to verify the effectiveness of our method, we first construct a **strong baseline** on Post-training Quantization Winograd. Concretely, we resort to Adaround [18] and LSQ [14] to quantize $U$ and $V$, respectively. Following BRECQ [19], we use 1024 unlabeled images and Adam [39] optimizer with 20k iterations and a batch size of 32. Experiments show that our strong baseline has surpassed previous state-of-the-art PTQ Winograd work. Based on it, we further verify our method on the strong baseline. The learning rates of $A$, $B$, and $G$ are set to (1e-4, 1e-4, 5e-4) by default, and the reason will be shown in Section 5.4. All our experiments are conducted on NVIDIA GeForce RTX 3090 24GB GPU servers and last for several hours.

### 5.2 PTQ-Aware Winograd

In this section, we compare our PTQ-Aware Winograd method to the previous work BQW [24] with comprehensive experiments settings, including various bitwidths, tile sizes, datasets and models.

Table 4: PTQ results of ResNet-20 on CIFAR-10 with different tile sizes (4,6) and different bitwidths (4,6,8).

| Model | Winograd Algorithm | Bits | Quantization Accuracy | | |
|---|---|---|---|---|---|
| | | | Strong Baseline | BQW [24] | **PAW** |
| | | 4 | 22.46 | 17.33 | **62.50 (+45.17)** |
| | F(4,3) | 6 | 80.63 | 79.36 | **90.25 (+12.65)** |
| ResNet-20 | | 8 | 89.55 | 90.31 | **92.02 (+1.71)** |
| (91.76%) | | 4 | 13.51 | 10.00 | **21.71 (+11.71)** |
| | F(6,3) | 6 | 46.62 | 39.75 | **85.29 (+45.54)** |
| | | 8 | 81.44 | 81.44 | **91.10 (+9.66)** |

**CIFAR-10**. The results of different PTQ methods on ResNet-20 are shown in Table 4. Although our strong baseline has been able to outperform previous method, our PTQ-Aware Winograd (PAW) can even improve it further, especially with lower bitwidth or larger tile size. For example, our methods surpass BQW [24] by 12.65 % and 45.54% when using 6-bit quantization on F(4,3) and F(6,3), respectively, which means our method makes deploying CNNs with more limited computation resources possible.

**ImageNet**. We also conduct experiments on ImageNet, a more challenging image classification dataset than CIFAR-10. In this experiment, we replace the convolutions in the last block with F(4,3) Winograd convolution since F(6,3) Winograd convolution needs more padding for $7 \times 7$ input size. Noting that BQW [24] only conducts PTQ with 8-bit because of severe accuracy degeneration on lower bitwidth, we compare our method with our strong baseline. The results presented in Table 5 demonstrate two breakthroughs of our method. First, we are the first to achieve negligible accuracy drop with 8-bit post-training quantization on Winograd convolution. Second, we are the first to achieve an acceptable accuracy drop with 6-bit post-training quantization on Winograd convolution.

## 5.3 Factorized Scale Quantization

In this section, we use Factorized Scale Quantization (FSQ) for $O$ and per-pixel quantization for $U$ and $V$. Other components are conducted with per-tensor quantization. We conduct experiments on ImageNet with bitwidths 6 and 8. In 6-bit quantization, we use 6-bit to quantize $U$, $V$, and 8-bit to quantize other components. The results shown in Table 5 indicate that our proposed FSQ is the crucial step to fully quantizing Winograd convolutions, and it is well-compatible with PTQ-Aware Winograd (PAW). Even with fully-quantization settings, our methods can achieve comparable accuracy compared to BQW[24].

Table 5: PTQ results of ResNets on ImageNet. Previous method BQW [24] does not quantize matrix multiplications in Winograd transformation. 'SB' indicates our Strong Baseline.

| Model | Tile | Bits | Partial Quantization | | | Full Quantization | | |
|---|---|---|---|---|---|---|---|---|
| | | | SB | BQW [24] | **PAW** | PAW | FSQ | **FSQ+PAW** |
| ResNet-18 (69.76%) | F(4,3) | 6 | 50.86 | N/A | **65.15** | 0.26 | 44.21 | **64.34** |
| | | 8 | 68.08 | 67.54 | **69.06** | 0.21 | 64.46 | **68.16** |
| | F(6,3) | 6 | 24.58 | N/A | **59.58** | 0.13 | 18.68 | **59.30** |
| | | 8 | 65.84 | 60.09 | **68.36** | 0.13 | 58.48 | **66.89** |
| ResNet-34 (73.30%) | F(4,3) | 6 | 58.85 | N/A | **69.71** | 0.88 | 52.10 | **68.80** |
| | | 8 | 72.11 | 71.86 | **72.67** | 0.96 | 69.51 | **71.75** |
| | F(6,3) | 6 | 27.91 | N/A | **63.56** | 0.19 | 22.06 | **62.11** |
| | | 8 | 69.68 | 66.52 | **71.90** | 0.29 | 63.60 | **69.72** |
| ResNet-50 (76.15%) | F(4,3) | 6 | 73.25 | N/A | **74.94** | 22.44 | 72.21 | **74.75** |
| | | 8 | 75.89 | 75.84 | **76.01** | 17.80 | 75.90 | **75.74** |
| | F(6,3) | 6 | 69.20 | N/A | **73.99** | 5.97 | 66.85 | **73.84** |
| | | 8 | 75.34 | 74.47 | **75.80** | 3.20 | 73.66 | **75.36** |

## 5.4 Ablation studies

To evaluate the effectiveness of our quantization-aware Winograd, we conduct an ablation study on ResNet-18 by optimizing different combinations of $A$, $B$, and $G$. Empirically, we find that the learning rate of $G$ is more sensitive than those of $A$ and $B$. Thus we fix the learning rates of $A$ and $B$ to 1e-4 for all experiments and perform a grid search on the learning rate of $G$ separately, which is varied in the interval [1e-7, 1e-3]. The best results are shown in Table 6. When optimizing them, the optimal learning rate for $G$ is 5e-4, and the performance improvement reaches 1.22%.

Table 6: Impact of different design choices for optimization of transformation matrices, on the ImageNet validation accuracy (%) for ResNet-18.

| Optimized Matrices | None | A | B | G | A, B | B, G | A, G | A, B, G |
|---|---|---|---|---|---|---|---|---|
| Accuracy | 68.04 | 68.07 | 68.24 | 68.03 | 68.87 | 68.28 | 68.03 | 69.26 |

## 6 Conclusions

This paper focuses on accelerating deep convolution neural networks by combining the Winograd algorithm and model quantization. We propose a unified optimization procedure to improve the compatibility between quantization and the Winograd algorithm. To achieve a more significant speed-up via full quantization, we propose a theoretically supported and hardware-friendly method called Factorized-Scale Quantization, which is suitable for the distribution characteristics of tensors in the Winograd domain. Experiments are conducted to show that our method can boost performance by a large margin, especially with lower bitwidth or larger tile size, which means our method makes deploying CNNs with more limited computation resources possible.

## 7 Acknowledgements

This work was supported in part by the STI 2030-Major Projects (No.2021ZD0201504), the NSFC-General Technology Collaborative Fund for Basic Research (No.U1936204), the Jiangsu Key Research and Development Plan (No.BE2023016, No.BE2021012-2).

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

# A Winograd transformation matrices

Depending on the particular choice of Winograd domain (i.e., polynomial domain), transformation matrices $A$, $B$, and $G$ in the Winograd algorithm can be different. In the paper, we present that the most popular interpolation points for F(2,3) are $[0, +1, -1]$ and then these transformation matrices can be constructed as follows:

$$A^T = \begin{bmatrix} 1 & 1 & 1 & 0 \\ 0 & 1 & -1 & -1 \end{bmatrix}, \quad B^T = \begin{bmatrix} 1 & 0 & -1 & 0 \\ 0 & 1 & 1 & 0 \\ 0 & -1 & 1 & 0 \\ 0 & 1 & 0 & -1 \end{bmatrix}, \quad G = \begin{bmatrix} 1 & 0 & 0 \\ \frac{1}{2} & \frac{1}{2} & \frac{1}{2} \\ \frac{1}{2} & -\frac{1}{2} & \frac{1}{2} \\ 0 & 0 & 1 \end{bmatrix} \tag{30}$$

For F(4,3) and F(6,3), we choose the same transformation matrices as BQW [24]. For F(4,3), the Winograd transformation matrices are as follows:

$$A^T = \begin{bmatrix} 1 & 1 & 1 & 1 & 1 & 0 \\ 0 & 1 & -1 & 2 & -2 & 0 \\ 0 & 1 & 1 & 4 & 4 & 0 \\ 0 & 1 & -1 & 8 & -8 & 1 \end{bmatrix}, \tag{31}$$

$$B^T = \begin{bmatrix} 4 & 0 & -5 & 0 & 1 & 0 \\ 0 & -4 & -4 & 1 & 1 & 0 \\ 0 & 4 & -4 & -1 & 1 & 0 \\ 0 & -2 & -1 & 2 & 1 & 0 \\ 0 & 2 & -1 & -2 & 1 & 0 \\ 0 & 4 & 0 & -5 & 0 & 1 \end{bmatrix}, \tag{32}$$

$$G = \begin{bmatrix} \frac{1}{4} & 0 & 0 \\ -\frac{1}{6} & -\frac{1}{6} & -\frac{1}{6} \\ -\frac{1}{6} & \frac{1}{6} & -\frac{1}{6} \\ \frac{1}{24} & \frac{1}{12} & -\frac{1}{6} \\ \frac{1}{24} & -\frac{1}{12} & -\frac{1}{6} \\ 0 & 0 & 1 \end{bmatrix} \tag{33}$$

For F(6,3), the Winograd transformation matrices are as follows:

$$A^T = \begin{bmatrix} 1 & 1 & 1 & 1 & 1 & 0 \\ 0 & 1 & -1 & 2 & -2 & 0 \\ 0 & 1 & 4 & 4 & 0 \\ 0 & 1 & -1 & 8 & -8 & 1 \end{bmatrix}, \tag{34}$$

$$B^T = \begin{bmatrix} 1 & 0 & -\frac{21}{4} & 0 & \frac{21}{4} & 0 & -1 & 0 \\ 0 & 1 & 1 & -\frac{17}{4} & -\frac{17}{4} & 1 & 1 & 0 \\ 0 & -1 & 1 & \frac{17}{4} & -\frac{17}{4} & -1 & 1 & 0 \\ 0 & \frac{1}{2} & \frac{1}{4} & -\frac{5}{2} & -\frac{5}{4} & 2 & 1 & 0 \\ 0 & -\frac{1}{2} & \frac{1}{4} & \frac{5}{2} & -\frac{5}{4} & -2 & 1 & 0 \\ 0 & 2 & 4 & -\frac{5}{2} & -5 & \frac{1}{2} & 1 & 0 \\ 0 & -2 & 4 & \frac{5}{2} & -5 & -\frac{1}{2} & 1 & 0 \\ 0 & -1 & 0 & \frac{21}{4} & 0 & -\frac{21}{4} & 0 & 1 \end{bmatrix}, \tag{35}$$

$$G = \begin{bmatrix} 1 & 0 & 0 \\ -\frac{2}{9} & -\frac{2}{9} & -\frac{2}{9} \\ -\frac{2}{9} & \frac{2}{9} & -\frac{2}{9} \\ \frac{1}{90} & \frac{1}{45} & -\frac{2}{45} \\ \frac{1}{90} & -\frac{1}{45} & \frac{2}{45} \\ \frac{32}{45} & \frac{16}{45} & \frac{8}{45} \\ \frac{32}{45} & -\frac{16}{45} & \frac{8}{45} \\ 0 & 0 & 1 \end{bmatrix} \tag{36}$$

# B Derivatives of transformation matrices

In the paper, in order to align these transformation procedures after quantization, we propose to adjust transformation matrices via an optimization procedure as follows:

$$\underset{A,B,G}{argmin} \quad \mathbb{E}_{X \sim \mathcal{D}} \left[ \Sigma_f^{C_o} ||A^T(\Sigma_c^{C_i} Q(B^T X_c B) \odot Q(GW_{f,c}G^T))A - Y_f||^2 \right] \tag{37}$$

By using the straight-through estimator [31] to approximate the gradient through the round function as a pass-through operation, we can obtain the derivatives of $A$, $B$ and $G$. In this paper, we directly present the derivative of $B$. Here, a more comprehensive derivation is provided as follows:

$$\frac{\partial \mathcal{L}}{\partial B_{ij}} = \Sigma_f^{C_o} \ \text{tr} \left\{ \frac{\partial \mathcal{L}}{\partial O_f^T} \cdot \frac{\partial O_f}{\partial B_{ij}} \right\} \tag{38}$$

$$= \Sigma_f^{C_o} \ \text{tr} \left\{ \frac{\partial \mathcal{L}}{\partial O_f^T} \cdot \left[ \Sigma_c^{C_i} (\delta_{ji} X_c B) \odot Q(V_{f,c}) + (B^T X_c \delta_{i,j}) \odot Q(V_{f,c}) \right] \right\} \tag{39}$$

$$= \Sigma_f^{C_o} \ \Sigma_c^{C_i} \ \text{tr} \left\{ \frac{\partial \mathcal{L}}{\partial O_f^T} \cdot \left[ (\delta_{ji} X_c B) \odot Q(V_{f,c}) \right] + \frac{\partial \mathcal{L}}{\partial O_f^T} \cdot \left[ (B^T X_c \delta_{ij}) \odot Q(V_{f,c}) \right] \right\} \tag{40}$$

$$= \Sigma_f^{C_o} \ \Sigma_c^{C_i} \ \text{tr} \left\{ (\delta_{ji} X_c B)^T \cdot \left[ \frac{\partial \mathcal{L}}{\partial O_f} \odot Q(V_{f,c}) \right] + (B^T X_c \delta_{ij})^T \cdot \left[ \frac{\partial \mathcal{L}}{\partial O_f} \odot Q(V_{f,c}) \right] \right\} \tag{41}$$

$$= \Sigma_f^{C_o} \ \Sigma_c^{C_i} \ \left[ X_c B \cdot (\frac{\partial L}{\partial O_f} \odot Q(V_{f,c}))^T \right]_{ij} + \left[ X_c^T B \cdot (\frac{\partial L}{\partial O_f} \odot Q(V_{f,c})) \right]_{ij} \tag{42}$$

We have obtained the derivative of $B_{ij}$, and now we can provide the expression for the derivative of $B$:

$$\frac{\partial \mathcal{L}}{\partial B} = \Sigma_f^{C_o} \ \Sigma_c^{C_i} \ X_c B (\frac{\partial L}{\partial O_f} \odot Q(V_{f,c}))^T + X_c^T B (\frac{\partial L}{\partial O_f} \odot Q(V_{f,c})) \tag{43}$$

The derivatives of $A$, $G$ and $O_f$ can be computed in a similar manner:

$$\frac{\partial \mathcal{L}}{\partial A} = \Sigma_f^{C_o} \ O_f^T A (A^T O_f A - Y_f) + O_f A (A^T O_f A - Y_f)^T \tag{44}$$

$$\frac{\partial \mathcal{L}}{\partial G} = \Sigma_f^{C_o} \ \Sigma_c^{C_i} \ (\frac{\partial L}{\partial O_f} \odot Q(U_c))GW_{f,c}^T + (\frac{\partial L}{\partial O_f} \odot Q(U_c))^T GW_{f,c} \tag{45}$$

$$\frac{\partial \mathcal{L}}{\partial O_f} = 2A(A^T O_f A - Y)A^T \tag{46}$$

# C Optimal quantization scale for Guassion varibles

In Theorem 1, in order to demonstrate that the optimal per-pixel scale $S$ can be factorized into vectors, we rely on the conclusion that the optimal scale $s^*$ to minimize the mean-square error of quantization of Gaussian variables $z \sim \mathcal{N}(0, \sigma^2)$ is proportional to $\sigma$, i.e., $s^* = K\sigma$, where $K$ is a constant. Here, we will provide a proof of it.

**Theorem 2.** *Assuming $z \sim \mathcal{N}(0, \sigma^2)$, the optimal scale $s^*$ to minimize the mean-square error of quantization of $z$ is proportional to the standard deviation $\sigma$, i.e., $s^* = K\sigma$, where $K$ is a constant.*

*Proof.* Because $z \sim \mathcal{N}(0, \sigma^2)$, $z$ can be reparameterized as $z = \sigma \cdot u$, where $u \sim \mathcal{N}(0, 1)$.

$$\mathbf{E}\left[(Q(z)-z)^2\right] = \int_{-\infty}^{\infty} p_z(z)(Q(z)-z)^2 dz \tag{47}$$

$$= \int_{-\infty}^{\infty} p_u(u)(Q(\sigma u)-\sigma u)^2 du \tag{48}$$

$$= \int_{-\infty}^{\infty} p_u(u)(clip\left(\left\lfloor\frac{\sigma u}{s}\right\rceil, -q_{min}, q_{max}\right)\cdot s - \sigma u)^2 du \tag{49}$$

$$= \sigma^2 \int_{-\infty}^{\infty} p_u(u)(clip\left(\left\lfloor\frac{u}{s/\sigma}\right\rceil, -q_{min}, q_{max}\right)\cdot \frac{s}{\sigma} - u)^2 du \tag{50}$$

$$= \sigma^2 h(\frac{s}{\sigma}) \tag{51}$$

Eq. (47) can be treated as a function of $s/\sigma$ when solving for $s$ with $\sigma$ as a known value. Assuming $K$ minimizes function $h(x)$, i.e., $K = \underset{x}{argmin}\ h(x)$, we have:

$$s^* = \underset{s}{argmin}\ \mathbf{E}\left[(Q(z)-z)^2\right] = \underset{s}{argmin}\ \sigma^2 h(\frac{s}{\sigma}) = K\cdot\sigma \tag{52}$$

$\square$

## D   Experiments on other architectures

In Section 5, we compare our methods to previous work BQW[24] on the ResNet model family with comprehensive experiment settings, including various bit widths, tile sizes, and datasets. Here, we present a similar analysis for two other popular architectures VGG and Squeezenet using the Cifar-10 dataset. The results are shown in Table 1 and Table 2. These results align with our analysis in Section 5. Our PTQ-Aware Winograd (PAW) method outperforms the strong baseline introduced in Section 5 and our FSQ method is well-compatible with PAW.

Table 1: PTQ results of VGG11 on CIFAR-10.

| Model | Tile | Bits | Partial Quantization | | Full Quantization | |
|---|---|---|---|---|---|---|
| | | | Baseline | **PAW** | FSQ | **FSQ+PAW** |
| VGG-11 (92.02%) | F(4,3) | 6 | 89.13 | 91.56 | 86.59 | 91.55 |
| | | 8 | 92.02 | 92.28 | 90.82 | 91.83 |
| | F(6,3) | 6 | 75.10 | 89.94 | 68.98 | 90.34 |
| | | 8 | 91.27 | 91.88 | 88.44 | 91.63 |

Table 2: PTQ results of SqueezeNet on CIFAR-10.

| Model | Tile | Bits | Partial Quantization | | Full Quantization | |
|---|---|---|---|---|---|---|
| | | | Baseline | **PAW** | FSQ | **FSQ+PAW** |
| SqueezeNet (92.62%) | F(4,3) | 6 | 89.69 | 91.98 | 88.66 | 91.78 |
| | | 8 | 92.61 | 92.68 | 92.01 | 92.80 |
| | F(6,3) | 6 | 80.50 | 90.67 | 76.48 | 91.26 |
| | | 8 | 92.37 | 92.61 | 90.54 | 92.42 |

