# OpenReview forum: "Towards Efficient and Accurate Winograd Convolution via Full Quantization"
_NeurIPS.cc/2023/Conference — NeurIPS 2023 poster_

### Official Review · Reviewer_M3Gq · 2023-07-05

**Soundness:** 3 good
**Presentation:** 3 good
**Contribution:** 3 good
**Rating:** 5
**Confidence:** 3

**Summary:**

This paper proposed to fully quantize the Winograd convolution post-training under the observation of disruption of consistency between different transformation procedures, the new proposed Factorized Scale Quantization is suitable in the Winograd domain. The experiments demonstrate significant improvements compared with previous post-training-quantization methods.

**Strengths:**

1. This paper is organized well, the exploration of quantization of different components of Winograd convolution clearly shows the primary cause of accuracy degeneration, and the following proposed method precisely targets this problem.
2. The proposed method is interesting, which utilizes two factor vectors to replace the per-pixel scales, and these two vectors can be merged into the transformation matrices.
3. Some experimental results are even higher than the baseline.


**Weaknesses:**

The experiments only demonstrate the quantization bit, there is no computation cost or inference time comparison which are very important to this work, it is difficult to know how much the proposed improve the efficiency other than accuracy.

**Questions:**

This paper emphasizes the hardware-friendly deployment of this proposed quantization method, will the optimization procedure including solving $\alpha$, $\beta$, and $\tilde{O}$ cost much more additional resources?

**Limitations:**

This paper hasn't mentioned or discussed its limitations.

---

> ### Author Rebuttal · Authors · 2023-08-10
>
> **Q1:** The experiments only demonstrate the quantization bit, there is no computation cost or inference time comparison which are very important to this work, it is difficult to know how much the proposed improve the efficiency other than accuracy.
>
> **A1:** Thank you for your suggestion.  Because of the time limitation, we can't implement  Winograd convolution on GPUs or FPGAs, necessitating low-level optimizations. Instead, we opt for BOPs as an alternative metric for measuring computation cost. This metric is widely used in various fields such as Neural Architecture Search (NAS), pruning, and quantization research(\[4,5,6])
>
> BOPs is defined as $BOPs=b_1 \cdot b_2 \cdot MAC$, where $b_1$ and $b_2$ represent the bit-width of two operators, respectively. **Please notes that the optimization procedures (13) and (26) happen before inference.** During the inference process, the computation cost of quantized Winograd convolution comprises three components: **element-wise multiplications** $U\odot V$, **Winograd transformations ($BXB^T$ and $AOA^T$)**, and **quantization overhead**.
>
> For element-wise multiplication, like [1, 3], our methods use the per-pixel quantization, with the memory overhead of  $(m+r-1)*(m+r-1)$ to store the quantization scales and $N\times C_{out}\times (m+r-1)\times (m+r-1)+ C_{in}\times C_{out}\times (m+r-1)\times(m+r-1)$ times flops to re-quantize $U$ and $V$.
>
> For Winograd transformations ($BXB^T$ and $AOA^T$), as introduced in Section 4.2.2, the benefit of our proposed FSQ is that we can move scales $\alpha$ and $\beta$ into transformation matrices. **So we can facilitate per-tensor matrix multiplication implementation when per-pixel quantization is utilized**. Therefore, quantizing $X$, $B^TX$, $O$, and $A^TO$ results in a constant memory overhead and a computation overhead of $2 \times N \times (C_{out} + C_{in}) \times (m + r - 1) \times (m + r - 1)$. Because these quantization and re-quantization operations need the same times flops as the tensor size, the overhead is negligible compared to Winograd transformations which involve twice matrix multiplications.
>
> The BOPs of different methods to quantize F(6,3) ResNet-20 are shown in Table R5-1. [1] quantizes all the transformation matrices $A$,$B$, and $G$, but the intermediate results such as $B^TX$ and $A^TO$ are not quantized. So [1] needs higher precision to hold these results and carry out the next operation. [2] don't quantize these Winograd transformations to maintain accuracy. Compared to them, our full-quantization of Winograd will achieve less computation cost.
>
> **Table R5-1 BOPs**
>
> |                    | Im2col(FP) | Winograd(FP) | BQW[2]  | Winograd-AwareNet[1] | Ours(PAW) | Ours(PAW+FSQ) |
> | ------------------ | ---------- | ------------ | ------- | -------------------- | --------- | ------------- |
> | Low-precision BOPs | 0          | 0            | 464.56M | 1282.45M             | 464.56M   | 791.71M       |
> | Flops              | 40.81M     | 12.37M       | 5.27M   | 0.48M                | 5.27M     | 0.80M         |
> | Total BOPs         | 10.44G     | 3.16G        | 1.81G   | 1.41G                | 1.81G     | 0.99G         |
>
> **Q2:** This paper emphasizes the hardware-friendly deployment of this proposed quantization method, will the optimization procedure including solving $\alpha$, $\beta$, and $O$ cost much more additional resources?
>
> **A2:** The overhead of the optimization procedure to solve $\alpha$, $\beta$, and $O$ is negligible. There are two reasons: (1) The optimization procedure (13) and (26) happens before inference.  (2) According to our experiments, this optimization procedure (26) converges fast. In most cases, it will converge in several to dozens of iterations, which takes approximately several minutes for one layer.
>
>
>
> [1] Fernández-Marqués et al., "Searching for Winograd-aware Quantized Networks", 2020
>
> [2] Chikin et al.,  "Channel Balancing for Accurate Quantization of Winograd Convolutions", 2022
>
> [3] Andir et al., "Going Further With Winograd Convolutions:Tap-Wise Quantization for Efficient Inference on 4x4 Tiles", 2022
>
> [4] Wang et al.,  "Differentiable Joint Pruning and Quantization for Hardware Efficiency", 2020
>
> [5] Guo et al.,  "Single path oneshot neural architecture search with uniform sampling", 2020
>
> [6] Liu et al.,  "Towards precise binary neural network with generalized activation functions", 2020

---

### Official Review · Reviewer_BWC3 · 2023-07-06

**Soundness:** 3 good
**Presentation:** 3 good
**Contribution:** 3 good
**Rating:** 6
**Confidence:** 3

**Summary:**

This paper proposes PTQ-Aware Winograd (PAW), Factorized-scale quantization and a iterative optimization algorithm to solve the problem of quantization on Winograd domain. These methods not only fully quantize the whole Winograd Convolution, but also surpass the existing Winograd quantization methods in terms of effect and hardware friendliness.

**Strengths:**

1. This is the first work to perform full-quantization for Winograd Convolution

2. Factorized-scalequantization makes the quantization for Winograd more hardware-friendly compared to per-pixel quantization in previous work

3. This paper focuses on the properties of the Winograd domain that make quantification difficult and gives a full analysis

4. The effect of the method is obviously beyond the existing methods, and the theoretical derivation is complete and the experiment is fully credible



**Weaknesses:**

1. In the explanation in Section 4.1:

   - Is the motivation of this section: In the original Winograd method, A, B and G satisfy some strict mathematical relationship with each other, but the perturbation brought by quantization destroys the strict relationship between them, and the small perturbation brought by such quantization will bring much greater overall loss. Therefore, the authors simulated the perturbation by artificially adding the perturbation and tested the Reconstruction Loss?

   - How was the range of perturbation added determined, and was the actual quantization error measured and used as a reference? Have you tested how much the Reconstruction Loss will decrease after optimization?

2. This work achieves an obvious improvement over previous work, but what are the challenges if we continue to expand the tile size? Why?

I would like to hear the authors’ feedback during rebuttal, and fixing these issues would further improve the quality of this paper.



**Questions:**

See Weakness.

**Limitations:**

See Weakness.

---

> ### Author Rebuttal · Authors · 2023-08-10
>
> **Q1:** In the explanation in Section 4.1:
>
> Is the motivation of this section: In the original Winograd method, A, B, and G satisfy some strict mathematical relationship with each other, but the perturbation brought by quantization destroys the strict relationship between them, and the small perturbation brought by such quantization will bring much greater overall loss. Therefore, the authors simulated the perturbation by artificially adding the perturbation and tested the Reconstruction Loss?
>
> How was the range of perturbation added determined, and was the actual quantization error measured and used as a reference? Have you tested how much the Reconstruction Loss will decrease after optimization?
>
> **A1:** (a) Your understanding of the motivation is right, but  your analysis is a little different from ours.
>
> In the original Winograd method, $A$, $B$, and $G$ satisfy some strict mathematical relationships with each other to ensure these Winograd transformations happen in the same Winograd domain. However, quantization noise destroys the relationship between them. For example, the weight transformation after quantization becomes $Q(GWG^T)$ , which may mismatch with original transformations $AOA^T$ and $B^TXB$.
>
> To demonstrate this, we change transformation matrices $A$ and $B$ towards different directions (by adding random noise) and test the reconstruction loss using $Q(GWG^T)$ and $GWG^T$, respectively. In Figure 1,  we discover that although original A and B can produce desired results with floating-point $GWGT$, they are no longer optimal when changing $GWG^T$ to $Q(GWG^T)$. About half of the random matrices A and B lead to more minor errors (the blue dots on the upper left of the yellow triangle). Thus, it is necessary to align these transformation procedures after quantization.
>
> (b) We test the reconstruction loss decreasing before and after optimization. Typically, the reconstruction loss will decrease to about 1/3 of the original loss. We show the results of different layers on ResNet-18 in Table R4-1.
>
> **Table R4-1. Reconstruction loss **
>
> |        | layer1.0.conv2 | layer2.0.conv2 | layer3.0.conv2 | layer4.0.conv2 |
> | ------ | -------------- | -------------- | -------------- | -------------- |
> | Before | $7.21e-02$     | $1.56e-01$     | $4.71e-01$     | $2.65e-01$     |
> | After  | $1.73e-02$     | $6.17e-02$     | $1.27e-01$     | $6.92e-02$     |
>
> **Q2:** This work achieves an obvious improvement over previous work, but what are the challenges if we continue to expand the tile size? Why?
>
> **A2:** We don't expand our methods for tile size >=8 in the paper because of two reasons:
>
> On the one hand, The speedup ratio of Winograd convolution,  represented by $\frac{(mr)^2}{(m+r-1)^2}$, increases more slowly as the tile size becomes larger. As an illustration, the speedup ratios for $F(2,3)$, $F(4,3)$, $F(6,3)$, and $F(8,3)$ are 2.25, 4, 5.0625, and 5.76, respectively (Table R4-2). On the other hand, Winograd convolution suffers from numeric accuracy issues: the floating point (FP) error increases exponentially with the tile size [1]. According to [1], the normalized L1 norm of floating point error for various deep convolution neural networks is shown in Table R4-2. Therefore, considering the trade-off between accuracy and efficiency, most papers [2,3,4] focus on F(4,3) and F(6,3).
>
>  **Table R4-2 Numerical Error and Speedup**
>
> |                     |   2    | 4                     | 6                | 8                | 10               |
> | ------------------- | :----: | --------------------- | ---------------- | ---------------- | ---------------- |
> | ResNet-20 FP error  | -----  | $1.19 \times 10^{−3}$ | $3.23 × 10^{−3}$ | $7.46 × 10^{−3}$ | $7.88 × 10^{−2}$ |
> | SqueezeNet FP error | ------ | $7.31\times 10^{-5}$  | $1.32 × 10^{−4}$ | $2.17 × 10^{−4}$ | $1.25 × 10^{−3}$ |
> | Speedup             |  2.25  | 4                     | 5.0625           | 5.76             | 6.25             |
>
> [1] Alam et al., "Winograd Convolution for Deep Neural Networks : Efficient Point Selection", 2022
>
> [2] Chikin et al.,  "Channel Balancing for Accurate Quantization of Winograd Convolutions", 2022
>
> [3] Andir et al., "Going Further With Winograd Convolutions:Tap-Wise Quantization for Efficient Inference on 4x4 Tiles", 2022
>
> [4] Fernández-Marqués et al., "Searching for Winograd-aware Quantized Networks", 2020

---

> > ### Comment · Reviewer_BWC3 · 2023-08-18
> >
> > Thank the authors for the detailed rebuttal, which solved my concerns. I believe it is a good work with constructive contributions to the field.

---

### Official Review · Reviewer_2v14 · 2023-07-11

**Soundness:** 3 good
**Presentation:** 3 good
**Contribution:** 3 good
**Rating:** 7
**Confidence:** 5

**Summary:**

The paper proposes a PTQ-Aware Winograd (PAW) method to improve the performance of deep learning inference with quantized parameters and Winograd Convolution. In particular, all steps of the Winograd operation are combined and optimized with a unified objective to reduce the domino effect of quantization in different parts of the Winograd sequence. Moreover, to mitigate the range difference of the Winograd output transformation, factorized-scale quantization is proposed. This involves balancing the distribution of the output transformation using two factorized scaling parameters. This approach outperforms previous work and the cifar-10 dataset.

**Strengths:**

The analysis of quantization for all parts of the Winograd operation is novel and interesting. The paper is well-written, and the proofs are explained well.

**Weaknesses:**

1- The overhead of proposed quantization algorithm needs to be compared with the QAT approach, since both methods perform training during the quantization process.

2- The new approach has only been tested on the ResNet model. It is suggested that experimental results for other networks be added to the paper.

**Questions:**

Most of the proofs in this paper assume the parameters follow a Gaussian distribution. Previous studies have shown that the distribution of parameters is similar to Laplace distribution. Does the proof in the paper also hold valid with a near-Laplace distribution of parameters?

**Limitations:**

The author does not discuss whether the new approach works for size 6 and size 8 of the Winograd operation.

---

> ### Author Rebuttal · Authors · 2023-08-10
>
> **Q1:** The overhead of proposed quantization algorithm needs to be compared with the QAT approach, since both methods perform training during the quantization process.
>
> **A1:**  Thank you for your suggestion. QAT methods require much more GPU resources and training data than PTQ. **When applying it to the Winograd algorithm,  these drawbacks will be amplified.** Training large Winograd models can be slow and use a significant amount of memory. This is a direct consequence of implementing all the stages involved in the Winograd convolution and retaining the inputs to each operation in memory to backpropagate through the entire process. According to the open source code of [1], the Winograd-aware QAT method costs about 10 hours on ResNet18 using the ImageNet dataset with **4 GPUs** (NVIDIA RTX 3090) for **one training epoch**. In contrast, the whole optimization procedure of the proposed PTQ method can be accomplished in around 5 hours on **a single GPU**.
>
> **The optimization procedures (13) and (26) happen before inference.**  During the inference process, as introduced in Section 4.2.2, the benefit of our proposed FSQ is that we can move scales $\alpha$ and $\beta$ into transformation matrices. So we can facilitate per-tensor matrix multiplication implementation when per-pixel quantization is utilized. The overhead of our proposed algorithm only includes re-quantization operation, which is negligible compared to the original Winograd transformation.
>
> **Q2:** The new approach has only been tested on the ResNet model. It is suggested that experimental results for other networks be added to the paper.
>
> **A2:**  Thank you for your suggestion. In the paper, we show the results of ResNet models to compare our method with another PTQ work [2]. Here, we add experiments on VGG and SqueezeNet using the Cifar-10 dataset. The results align with our expectations. The detailed results are presented in Table R3-1 and Table R3-2.
>
> **Table R3-1. Accuracy (\%) on VGG11 (92.02%).**
>
> | Tile Size | BRECQ[4] | PAW   | FSQ   | FSQ+PAW |
> | --------- | -------- | ----- | ----- | ------- |
> | F4        | 89.13    | 91.56 | 86.59 | 91.55   |
> |           | 92.02    | 92.28 | 90.82 | 91.83   |
> | F6        | 75.10    | 89.94 | 68.98 | 90.34   |
> |           | 91.27    | 91.88 | 88.44 | 91.63   |
>
> **Table R3-2. Accuracy (\%) on SqueezeNet (92.62%).**
>
> | Tile Size | BRECQ[4] | PAW   | FSQ   | FSQ+PAW |
> | --------- | -------- | ----- | ----- | ------- |
> | F4        | 89.69    | 91.98 | 88.66 | 91.78   |
> |           | 92.61    | 92.68 | 92.01 | 92.80   |
> | F6        | 80.50    | 90.67 | 76.48 | 91.26   |
> |           | 92.37    | 92.61 | 90.54 | 92.42   |
>
> **Q3:** Most of the proofs in this paper assume the parameters follow a Gaussian distribution. Previous studies have shown that the distribution of parameters is similar to Laplace distribution. Does the proof in the paper also hold valid with a near-Laplace distribution of parameters?
>
> **A3:**  **Yes, our proof is also valid for the near-Laplace distribution of parameters**. Actually, in Proposition 1, we only use the condition that $X$ and $W$ are zero-mean independent and identically distributed to calculate the mean and standard deviation of $O$. Therefore, the assumption that the parameters follow a Gaussian distribution is not necessary, and the proof also applies to the case of a Laplace distribution with mean zero (even other zero-mean distributions). We have modified the theorem in the revision.
>
> **Q4:** The author does not discuss whether the new approach works for size 6 and size 8 of the Winograd operation.
>
> **A4:** Thank you for your suggestion. **The experiments on tile size 6 have been shown in Table 4 and Table 5.** We don't expand our methods for tile size >=8 in the paper because of two reasons:  On the one hand, The speedup ratio of Winograd convolution,  represented by $\frac{(mr)^2}{(m+r-1)^2}$, increases more slowly as the tile size becomes larger. On the other hand, Winograd convolution suffers from numeric accuracy issues: the floating point (FP) error increases exponentially with the tile size [4]. According to [4], the normalized L1 norm of floating point error for various deep convolution neural networks is shown in Table R3-3. Therefore, considering the trade-off between accuracy and efficiency, most papers [2,3] focus on F(4,3) and F(6,3).
>
>  **Table R3-3 Numerical Error and Speedup**
>
> |                     |   2    | 4                     | 6                | 8                | 10               |
> | ------------------- | :----: | --------------------- | ---------------- | ---------------- | ---------------- |
> | ResNet-20 FP error  | -----  | $1.19 \times 10^{−3}$ | $3.23 × 10^{−3}$ | $7.46 × 10^{−3}$ | $7.88 × 10^{−2}$ |
> | SqueezeNet FP error | ------ | $7.31\times 10^{-5}$  | $1.32 × 10^{−4}$ | $2.17 × 10^{−4}$ | $1.25 × 10^{−3}$ |
> | Speedup             |  2.25  | 4                     | 5.0625           | 5.76             | 6.25             |
>
> [1] Fernández-Marqués et al,."Searching for Winograd-aware Quantized Networks", 2020
>
> [2] Chikin et al.,  "Channel Balancing for Accurate Quantization of Winograd Convolutions", 2022
>
> [3] Li et al., "BRECQ: Pushing the Limit of Post-Training Quantization by Block Reconstruction", 2021
>
> [4] Alam et al., "Winograd Convolution for Deep Neural Networks : Efficient Point Selection", 2022

---

> > ### Comment · Reviewer_2v14 · 2023-08-21
> > **Rebuttal Response.**
> >
> > Thank the authors for the detailed response to my comments. The responses are valid and complete. I also believe this is a novel and interesting paper, and I am raising my score to 7.

---

### Official Review · Reviewer_JmCw · 2023-07-25

**Soundness:** 3 good
**Presentation:** 4 excellent
**Contribution:** 3 good
**Rating:** 5
**Confidence:** 3

**Summary:**

This paper proposes a post-training quantization algorithm for Winograd convolution, which overcomes the inconsistency in domain transformation by adjusting the transformation matrices together (PTQ-Aware) via a unified optimization procedure, and achieves full quantization by a new factorized scale quantization (FSQ) method. Experiments on CIFAR-10 and ImageNet show the effectiveness of the proposed algorithm. It surpasses the previous state-of-the-art Winograd post-training quantization algorithm significantly in terms of accuracy.


**Strengths:**

* The proposed Winograd post-training quantization algorithm is considerably more accurate than previous methods, and is hardware friendly.
* The paper is well presented. I am able to understand the whole paper as a reader whose research focus is not on model quantization.

**Weaknesses:**

* The paper failed to mention or compare with some quantization-aware training (QAT) algorithms that achieve full-quantization, e.g "Searching for Winograd-aware Quantized Networks" [1] and "Going Further With Winograd Convolutions: Tap-Wise Quantization for Efficient Inference on 4x4 Tiles" [2]. These QAT algorithms share quite similar ideas with this paper and the accuracy of their final quantized models are much higher.
  * [1] https://arxiv.org/abs/2002.10711  (2020)
  * [2] https://arxiv.org/abs/2209.12982 (2022)
* If possible, some quantitive evaluation on how faster the proposed algorithm is (compared to previous algorithms) is desired.
* Two typos:
  * line 188: v_i -> v_j
  * line 239: long -> learn






**Questions:**

* Could you briefly compare your proposed algorithm with some QAT methods that also achieve full-quantization (e.g. the two papers I listed in the Weakness section) and summarize the advantage of your algorithm?
* Could you provide some evaluation on the inference speed of the proposed algorithm compared with previous algorithms? If not possible, some simple analysis is also OK. I'd like to know whether there's some hidden computation overhead or complexity introduced by your algorithm.


**Limitations:**

I hope the authors can better differentiate their method with previous Winograd quantization algorithms and justify their method's advantages. If it simply adapts QAT full-quantization to the PTQ scenario, the novelty and contribution of the paper would be somewhat undermined.

---

> ### Author Rebuttal · Authors · 2023-08-10
>
> **Q1**: Could you briefly compare your proposed algorithm with some QAT methods that also achieve full-quantization (e.g. the two papers I listed in the Weakness section) and summarize the advantage of your algorithm?
>
> **A1**: Thank you for your suggestion. This response includes three parts: **the difference between QAT and PTQ**, **a comparison with QAT algorithms**, and a **results comparison**.
>
> **In the quantization research community, QAT and PTQ are two distinct research areas.** Although QAT can achieve more promising performance, training the network requires great many GPU resources and a full training dataset. Therefore, it is not always practical when either the training dataset is unavailable (e.g., privacy and commercial training data) or rapid deployment is required. Therefore, much effort has been spent in the industry and research community toward improving PTQ performance.
>
> **The drawbacks of QAT will be amplified when it is applied to the Winograd algorithm.** Training large Winograd models can be slow and use a significant amount of memory. This is a direct consequence of implementing all the stages involved in the Winograd convolution and retaining the inputs to each operation in memory to backpropagate through the entire process. According to the open source code of [1], the Winograd-aware QAT method costs about 10 hours on ResNet18 using the ImageNet dataset with **4 GPUs** (NVIDIA RTX 3090) for **one training epoch**. In contrast, **the whole optimization procedure** of the proposed PTQ method can be accomplished in around 5 hours on **a single GPU**
>
> "Searching for Winograd-aware Quantized Networks" [1]  and "Going Further With Winograd Convolutions: Tap-Wise Quantization for Efficient Inference on 4x4 Tiles" are two classical papers about QAT for Winograd convolutions. **Compared to them,  we have the following difference and advantages**:
>
> -   [1] quantizes all the transformation matrices $A$,$B$, and $G$, but the intermediate results such as $B^TX$ and $A^TO$ are not quantized. So [1] needs higher precision to hold these results and carry out the next operation.  In addition, their methods, as mentioned above, require a full dataset and huge training resources, which are sometimes unavailable.
> -   [2] proposes tap-wise quantization for $U$ and $V$ which is used in many following works. They don't re-quantize the high-precision immediate results $O$ and utilize the shift-and-add operations to accelerate the computation of $A^TOA$.
>
> **Moreover, the strong effectiveness of retraining the whole network hides the difficulty of full quantization, especially for quantizing output tensors in the Winograd domain $O$**.  According to our analysis in Section 4.2.1,  $O$ suffers from huge distribution differences among taps and deserves per-pixel quantization (or called tap-wise quantization in [2]). However, since O will take part in matrix multiplication (Eq. 4) instead of element-wise multiplication, per-pixel quantization will lead to different scales in the summation dimension, which makes it not feasible in general hardware. Thus, we proposed a more hardware-friendly method FSQ, which factorizes per-pixel scales into two vectors and we can move both scales into transformation matrices. We also **theoretically** demonstrate that, under the assumption of the identical and independent distribution of weights and activations, our method is equally optimal to per-pixel quantization in minimizing quantization errors.
>
> It's also worth noting that although our methods only require few-shot, unlabeled calibrations, and fewer computation resources, **we can achieve comparable performance with previous QAT works.** The difference and the results of ResNet-20 using Cifar-10 dataset are shown in Table R2-1.
>
> **Table R2-1 Comparision with Other Methods**
>
> |                   | Quantization Type | Optimziation of Transformation Matrices             | Full Quantization                                       | Quantiation  graunlarity of $O$ | Acc drop    |
> | ----------------- | ----------------- | --------------------------------------------------- | ------------------------------------------------------- | ------------------------------- | ----------- |
> | BQW[3]            | QAT/PTQ           | Fixed                                               | Not quantize  Winograd transformation                   | Not Quantize                    | -0.02/-1.42 |
> | Winograd-Aware[1] | QAT               | End-to-end Training(Multiple GPUs and full dataset) | Not re-quantize the immediate results $A^TO$ and $B^TX$ | Per-tensor                      | -0.7        |
> | Tap-wise quant[2] | QAT               | Fixed                                               | Not re-quantize the immediate results $O$               | Not Quantize                    | -0.6        |
> | Ours              | PTQ               | Reconstruction Loss (Single GPU and unlabeled data) | Quantize all components                                 | FSQ                             | -0.47       |
>
> **Q2:** If possible, some quantitive evaluation on how faster the proposed algorithm  (compared to previous algorithms) is desired.
>
> **A2:** Because of the length limitation, we answer this question in the **global rebuttal**.
>
> **Q3:** Two typos: line 188: $v_i$ -> $v_j$ line 239: long -> learn
>
> **A3:** Thank you for your careful reading of the paper. These issues will be corrected in the updated manuscript.
>
> [1] Fernández-Marqués et al., "Searching for Winograd-aware Quantized Networks", 2020
>
> [2] Andir et al., "Going Further With Winograd Convolutions:Tap-Wise Quantization for Efficient Inference on 4x4 Tiles", 2022
>
> [3] Chikin et al.,  "Channel Balancing for Accurate Quantization of Winograd Convolutions", 2022

---

> > ### Comment · Reviewer_JmCw · 2023-08-20
> >
> > Thanks very much for the detailed rebuttal, which has solved most of my concerns. I am now leaning between "borderline accept" and "weak accept". But still, due to some concerns on the potential impact of the paper, I'd like to keep my current rating.

---

### Official Review · Reviewer_Rvu6 · 2023-07-26

**Soundness:** 3 good
**Presentation:** 2 fair
**Contribution:** 3 good
**Rating:** 5
**Confidence:** 3

**Summary:**

This paper presents a Post-training-quantization-aware Winograd (PAW) method to optimize all transformation procedures required by the Winograd algorithm to achieve post-training quantization of pre-trained ResNet models. A useful factorized scale quantization (FSQ) method is also proposed to balance the differences in the range of values in the Winograd domain. Obvious increases in terms of classification accuracy values on CIFAR-10 and ImageNet datasets are obtained with different ResNet-style model structures.

**Strengths:**

The paper achieved good improvements by solving problems observed in practice in cases with large tile sizes. It is useful to know the problems the authors observed and their solutions.

**Weaknesses:**

1. The presentation of the paper was not satisfactory and is quite difficult to follow. A lot of polishing work needs to be made before the paper could reach a state that I could recommend accepting.

2. Some motivation for the work needs to be justified: why checking and solving the issue happened when quantizing ResNet-style models with large (>=4) tile sizes? Some justification and performance differences need to be made when comparing to those with smaller tile sizes.

Many minor issues are listed below:
1. Many issues in the use of English:
a. "Another approach to accelerate CNNs is faster implementations of convolution, such as FFT [11] and Winograd [10]. Among them,"  ...
b. "for the first time. And we further propose a " ...
and many others (too many to list)...
2. "surpasses the previous state-of-the-art method by 8.27% and 5.38% on ResNet-18 and ResNet-34, respectively." -> Better to be clear surpass in terms of what?
3. Different numbers of significant figures are used in Table 2.
4. Abbreviation std is not defined in the paper.
5. "does not require retraining the network end-to-end." -> It seems the authors may mean both updating the entire model and training from scratch by using the wording "end-to-end" here. Many ambiguities like this exist throughout the paper.
6. Would be more friendly to the readers to define tile size in the paper (I guess you mean kernel size or filter size here). Could also provide the definitions of F(4,3) and F(6,3) before using them.
7. Many issues in the references. For instance, conference references [7], [19], and [30] are in different formats. Some references, such as [33], are not cited in their official publication sources.
8. In table 5, perhaps better to use the wording "baseline" than "strong baseline". We believe all baseline systems authors try to quote are strong enough.
9. Since non-standard tile sizes are used in this paper, it may be better to explain more carefully about the model architectures rather than simply referring to them using the standard terms (ResNet-34 etc.).

**Questions:**

1. From Table 4, the cases in which PAW resulted in the most considerable improvements are 6-bit and 8-bit for F(6,3). Could you please explain in what type of hardware, 6-bit could be a more efficient setting than 8-bit?

2. From Table 5, F(4,3) results are often better than the F(6,3) results, which matches my understanding. In that sense, could you please explain why improving the performance with tile size 6 matters? Following that logic, could you please provide more evidence of the advantage of using a large tile size (>=4)?

**Limitations:**

Only ResNet-style model architectures are tested, which is limited. The authors also did not compare the results and efficiency against those with the standard tile size.

---

> ### Author Rebuttal · Authors · 2023-08-10
>
> Thank you for your careful reading of our article and providing helpful feedback. We have scrutinized the manuscript and made corresponding modifications, e.g., correcting some issues in tables and references. We have also polished the paper per your recommendations and explained some specific terms (e.g., tile size) more clearly. **We apologize for the confusion between tile size and filter size.** The final version would be more reader-friendly, even for those unfamiliar with the Winograd algorithm. **We also analyze the relationship between speedup ratio and tile sizes and formally demonstrate the necessity of adopting large tile sizes to accelerate CNNs.**
>
> **Q1:** Some motivation for the work needs to be justified: why checking and solving the issue happened when quantizing ResNet-style models with large (>=4) tile sizes?
>
> **A1:**   We apologize for the confusion. The tile size $m$ in Winograd convolution differs from the filter size $r$ in standard convolution. **For $r \times r$ standard convolutions with filter size $r$, the algorithm transforms the convolution operations to the Winograd domain and generates $m \times m$ (spatial) outputs at a time, which is denoted as $F(m,r)$.** The parameter m is called tile size, which is used to balance the speedup and numerical precision.
>
> Compared with standard convolution, the Winograd algorithm requires fewer multiplications. The speedup ratio depends on tile size $m$. For a $m \times m$ output patch, $F(m,r)$ Winograd convolution requires $C_{in}C_{out}(m+r-1)^2+2C_{in}(m+r-1)^2+2C_{out}(m+r-1)^2m$ multiplications. In contrast, standard convolution requires $C_{in}C_{out}(mr)^2$ multiplications. Usually, filter numbers $C_{out}$ and channel numbers $C_{in}$ are much larger than $m$ and $r$ in modern CNNs, and the speedup ratio is about $\frac{(mr)^2}{(m+r-1)^2}$. Therefore, **the larger the tile size $m$, the more significant the speedup becomes (Table R1-1).**
>
> However, Winograd convolution suffers from numeric accuracy issues: the floating point (FP) error increases **exponentially** with the tile size [1] (Table R1-1). Therefore, considering the trade-off between accuracy and efficiency, most papers [5,7] focus on F(4,3) and F(6,3).
>
>  **Table R1-1 FP Error and Speedup**
>
> | |2|4|6|8|10|
> |-|:-:|-|-|-|-|
> |ResNet-20 FP error| --  |$1.19×10^{−3}$|$3.23×10^{−3}$|$7.46× 10^{−3}$|$7.88×10^{−2}$|
> |SqueezeNet FP error| --- |$7.31×10^{-5}$|$1.32×10^{−4}$|$2.17×10^{−4}$|$1.25×10^{−3}$|
> |Speedup|2.25|4|5.0625|5.76 |6.25|
>
> **Q2:** Many issues in the use of English: a. "Another approach to accelerate CNNs is faster implementations of convolution, such as FFT [11] and Winograd [10]. Among them," ... b. "for the first time. And we further propose a "
>
> **A2:** We have addressed the issues you highlighted as follows:
>
> (a) "An alternative method for enhancing the speed of CNNs involves the development of faster convolution implementations
>
> (b) "...for the first time. We further propose a..."
>
> (c) "...our proposed method outperforms ... in terms of the **top-1 accuracy** ".
>
> (d) " the standard deviation (**std**) of O distributed...."
>
> **Q3:** "Post-training quantization is a more lightweight method that does not require retraining the network end-to-end" -> It seems the authors may mean both updating the entire model and training from scratch by using the wording "end-to-end" here.
>
> **A3:** You are right. **QAT updates the entire model with end-to-end training, while PTQ only needs to optimize quantization parameters layer by layer.** Due to page constraints, we use the term "end-to-end" to convey this distinction in the related works section. Many other papers use the exact phrase to distinguish between PTQ and QAT, e.g., "A White Paper on Neural Network Quantization"[2].  In the final revision, we have modified the sentence as the bolded sentence.
>
> **Q4:** Since non-standard tile sizes are used in this paper, it may be better to explain more carefully about the model architectures rather than simply referring to them using the standard terms (ResNet-34 etc.).
>
> **A4:** As introduced in A2, the tile size is a parameter of the Winograd algorithm. The model architectures (e.g., filter size and input/output sizes of each layer) remain unchanged.
>
> **Q5:** Could you please explain in what type of hardware, 6-bit could be a more efficient setting than 8-bit?
>
> **A5:**  Although current general-purpose GPUs may not fully support 6-bit operations, many domain-specific accelerators can support precision-scalable computation. For example, Stripes[3] and BitFusion[4], both built on the bit-serial approach, can at run-time tune to the desired precision mode (e.g., 6-bit). In this type of hardware, the advantages of utilizing 6-bit computation over 8-bit computation are prominent and can be categorized into two key aspects:
>
> 1. **Reduced Storage Access and Energy Consumption:**
> 2. **Enhanced Computation Latency in Bit-Serial Scenarios:** In bit-serial architectures, 6-bit computation exhibits lower latency than 8-bit computation, achieving an acceleration effect of 1.3x [3].
>
> **Q6:** Only ResNet-style model architectures are tested, which is limited.
>
> **A6:** Because of the length limitation, we answer this question in the **global rebuttal.**
>
> [1] Alam et al., "Winograd Convolution for Deep Neural Networks : Efficient Point Selection", 2022
>
> [2] Nagel et al., "A White Paper on Neural Network Quantization", 2021
>
> [3] Judd et al., "Stripes**:** Bit-serial deep neural network computing", 2016
>
> [4] Sharma et al., "Bit Fusion: Bit-Level Dynamically Composable Architecture for Accelerating Deep Neural Network", 2018
>
> [5] Chikin et al., "Channel Balancing for Accurate Quantization of Winograd Convolutions", 2022
>
> [6] Li et al., "BRECQ: Pushing the Limit of Post-Training Quantization by Block Reconstruction", 2021
>
> [7]  Fernández-Marqués et al., "Searching for Winograd-aware Quantized Networks", 2020

---

### Author Rebuttal · Authors · 2023-08-10

Thank you reviewers for your helpful feedback and constructive advice. Based on the reviewers' questions, comments, and recommendations, we have made many revisions that may significantly improve the quality of the paper.

**Here, we explain the common concern of reviewers on the computation cost of our algorithm.**
 Because of the time limitation, we can't implement  Winograd convolution on GPUs or FPGAs, necessitating low-level optimizations. Instead, we opt for **BOPs** as an alternative metric for measuring computation cost. This metric is widely used in various fields such as Neural Architecture Search (NAS), pruning, and quantization research(\[1,2,3])

BOPs is defined as $BOPs=b_1 \cdot b_2 \cdot MAC$, where $b_1$ and $b_2$ represent the bit-width of two operators, respectively. **Please notes that the optimization procedures (13) and (26) happen before inference.** During the inference process, the computation cost of quantized Winograd convolution comprises three components: **element-wise multiplications** $U\odot V$, **Winograd transformations ($BXB^T$ and $AOA^T$)**, and **quantization overhead**.

For **element-wise multiplication**, like [1, 3], our methods use the per-pixel quantization, with the memory overhead of  $(m+r-1)*(m+r-1)$ to store the quantization scales and $N \times C_{out}\times (m+r-1)\times (m+r-1)+ C_{in}\times C_{out}\times (m+r-1)\times(m+r-1)$ times flops to re-quantize $U$ and $V$.

For **Winograd transformations ($BXB^T$ and $AOA^T$)**, as introduced in Section 4.2.2, the benefit of our proposed FSQ is that we can move scales $\alpha$ and $\beta$ into transformation matrices. **So we can facilitate per-tensor matrix multiplication implementation when per-pixel quantization is utilized**. Therefore, quantizing $X$, $B^TX$, $O$, and $A^TO$ results in a constant memory overhead and a computation overhead of $2 \times N \times (C_{in} + C_{out}) \times (m + r - 1) \times (m + r - 1)$. Because these quantization and re-quantization operations need the same times flops as the tensor size, the overhead is negligible compared to Winograd transformations which involve twice matrix multiplications.

The BOPs of different methods to quantize F(6,3) ResNet-20 are shown in Table R5-1. [1] quantizes all the transformation matrices $A$,$B$, and $G$, but the intermediate results such as $B^TX$ and $A^TO$ are not quantized. So [1] needs higher precision to hold these results and carry out the next operation. [2] don't quantize these Winograd transformations to maintain accuracy. Compared to them, our full-quantization of Winograd will achieve less computation cost.

**Table R1**

|                    | Im2col(FP) | Winograd(FP) | BQW[5]  | Winograd-AwareNet[4] | Ours(PAW) | Ours(PAW+FSQ) |
| ------------------ | ---------- | ------------ | ------- | -------------------- | --------- | ------------- |
| Low-precision BOPs | 0          | 0            | 464.56M | 1282.45M             | 464.56M   | 791.71M       |
| Flops              | 40.81M     | 12.37M       | 5.27M   | 0.48M                | 5.27M     | 0.80M         |
| Total BOPs         | 10.44G     | 3.16G        | 1.81G   | 1.41G                | 1.81G     | 0.99G         |


**Another common concern of reviewers is the performance of our algorithm on other networks.** In the paper, we show the results of ResNet-style models to compare our method with another PTQ work [5]. Here, we add experiments on VGG and Squeezenet using the Cifar-10 dataset. The results align with our expectations. The detailed results are presented in Table R1-2 and Table R1-3. We also add these experiments as Sec. 5 in supplementary materials.

**Table R2. Accuracy (\%) on VGG11 (92.02%).**

| Tile Size | BRECQ [6] | PAW   | FSQ   | FSQ+PAW |
| --------- | --------- | ----- | ----- | ------- |
| F4        | 89.13     | 91.56 | 86.59 | 91.55   |
|           | 92.02     | 92.28 | 90.82 | 91.83   |
| F6        | 75.10     | 89.94 | 68.98 | 90.34   |
|           | 91.27     | 91.88 | 88.44 | 91.63   |

**Table R3. Accuracy (\%) on SqueezeNet (92.62%).**

| Tile Size | BRECQ [6] | PAW   | FSQ   | FSQ+PAW |
| --------- | --------- | ----- | ----- | ------- |
| F4        | 89.69     | 91.98 | 88.66 | 91.78   |
|           | 92.61     | 92.68 | 92.01 | 92.80   |
| F6        | 80.50     | 90.67 | 76.48 | 91.26   |
|           | 92.37     | 92.61 | 90.54 | 92.42   |

[1] Wang et al.,  "Differentiable Joint Pruning and Quantization for Hardware Efficiency", 2020

[2] Guo et al.,  "Single path oneshot neural architecture search with uniform sampling", 2020

[3] Liu et al.,  "Towards precise binary neural network with generalized activation functions", 2020

[4] Fernández-Marqués et al., "Searching for Winograd-aware Quantized Networks", 2020

[5] Chikin et al.,  "Channel Balancing for Accurate Quantization of Winograd Convolutions", 2022

[6] Li et al., "BRECQ: Pushing the Limit of Post-Training Quantization by Block Reconstruction", 2021

---

### Author Response · Authors · 2023-08-19

Dear Reviewers,

We are wondering if our response has addressed all your concerns. As noted in our previous responses, we conducted additional experiments and attempted to address your concerns.  Thanks again for your valuable reviews.  If you have any further questions, please let us know.

With best regards,
The authors

---

### Decision · Program_Chairs · 2023-09-21

**Decision:**

Accept (poster)

**Comment:**

The paper introduces a Post-training-quantization-aware Winograd (PAW) method, aiming to optimize the transformation procedures required by the Winograd algorithm for post-training quantization of pre-trained ResNet models. The proposed method also includes a factorized scale quantization (FSQ) technique to address the range of values in the Winograd domain. The paper's findings are supported by experiments on CIFAR-10 and ImageNet datasets using various ResNet architectures.

The paper presents a novel approach to post-training quantization for Winograd convolution, which has been acknowledged by reviewers as a significant contribution. The proposed method demonstrates improvements over existing methods, especially in terms of classification accuracy. The paper's organization and structure have been appreciated, with some reviewers noting the clear presentation of the challenges and solutions in the Winograd domain.

The reviewers are all positive about the paper. Although some concerns are rasied, they are addressed in rebuttal. The AC agrees with the reviewers to accept the paper. The authors should focus on improving the paper's presentation and clarity, providing a more comprehensive experimental evaluation, and addressing the technical issues highlighted by the reviewers.